# Effects of black carbon morphology on brown carbon absorption estimation: from numerical aspects

Jie Luo, Yongming Zhang, and Qixing Zhang

State Key Laboratory of Fire Science, University of Science and Technology of China, Hefei, Anhui 230026, China

**Correspondence:** Qixing Zhang (qixing@ustc.edu.cn)

**Abstract.** In this work, we developed a numerical method to investigate the effects of black carbon (BC) morphology on the estimation of brown carbon (BrC) absorption using the Absorption Ångström exponent (AAE) methods. Pseudo measurements of the total absorption were generated based on several morphologically mixed BC models, then the BrC absorption was inferred based on different BC AAE methods. By investigating the estimated BrC absorption at different parameters, we have
demonstrated under what conditions the AAE methods can provide good/bad estimations. As recent studies have shown that both externally and internally mixed BC still exhibit a relatively small fractal dimension value, the AAE = 1 method is still a reasonable method to estimate the BrC absorption, as the AAE of fluffy BC does not deviate largely with 1. However, the deviation between the "True" and the estimated BrC mass absorption cross-section (MAC) should be also carefully considered, as sometimes the MAC deviation estimated using the AAE = 1 method can reach a value that is comparable to the "True"
BrC MAC for internally mixed particles. The Mie AAE method can just provide relatively reasonable estimations for small particles, and the BrC absorption deviations estimated using the Mie AAE methods are rather substantial for large particles. If the BC core still exhibits a fluffy structure, the deviation between the "True" and the estimated BrC MAC can reach 4.8 $m^2$/g and 5.8 $m^2$/g for large externally and internally mixed particles, respectively. Even for compact BC core, the BrC MAC deviation estimated using the Mie AAE method can reach approximately 2.8 $m^2$/g when the BC size is large. By comparing the
AAE of spherical BC and detailed BC models, we found that the AAE does not deviate largely with 1 if BC presents a fluffy fractal structure, while it varies largely with particle size if BC exhibits a spherical structure, and the AAE value of spherical BC can vary from a negative value to approximately 1.4. The WDA method does not necessarily improve the estimations. In many cases, the WDA method even provides a worse estimation than the BC AAE=1 and Mie AAE methods. Our results showed that the WDA does not deviate largely with 0 if the BC core presents a fluffy structure, while the WDA of spherical
BC can vary significantly as the particle size changes. The deviation between the "True" and the estimated BrC MAC using the WDA method can reach approximately 9 $m^2$/g for externally mixed particles, which is far more than BrC MAC itself. As recent studies have shown BC commonly exhibits a fluffy structure but not a spherical structure, the estimation of BrC absorption based on the AAE method should carefully consider the effects of BC morphologies.

# 1 Introduction

Carbonaceous aerosols, a main source of the light-absorbing aerosols, contribute great effects on the climate. Carbonaceous aerosols mainly include black carbon (BC) and organic carbon (OC). BC was considered as the dominant absorbing aerosol in the atmosphere, which greatly absorbs light from ultraviolet (UV) wavelengths to near-infrared wavelengths, and it contributes to large warming effects on the climate (Stocker et al., 2013). OC was often regarded as a scattering aerosol, while many studies have shown that parts of OC can also strongly absorb light in UV wavelengths (Kirchstetter et al., 2004; Chakrabarty

et al., 2010; Chen and Bond, 2010), and the absorbing OC is called brown carbon (BrC). To figure out the climate effects of BrC, many modeling studies have been studied. BrC was estimated to contribute to approximately 20 - 40% of the total aerosol absorption, and its direct radiative effect has been estimated to be comparable to that of BC (Feng et al., 2013; Saleh et al., 2015). However, substantial uncertainties exist in the climate modeling of BrC (Wang et al., 2016). The accurate estimation of BrC demands the constraints from the observation.

Laboratory measurements based on the extraction of filter samples were widely used to measure BrC absorption, while it is difficult to provide global, continuous measurements. Thus, increasing studies used measurements based on remote sensing and in-situ techniques. However, the observed absorptions commonly come from the mixing of different aerosols. To separate the contributions of different aerosols, some attempts were made to derive the BrC contribution from the total absorption (Wang et al., 2016, 2018; Russell et al., 2010; Massabò et al., 2015; Bahadur et al., 2012; Chung et al., 2012). Dust, BC, and BrC

are widely accepted to be the main absorbing aerosols in the atmosphere. Dust is recognized to be in the coarse mode, while BrC and BC are commonly in fine size mode. Therefore, based on the size information inferred from remote sensing using different techniques (eg. The extinction Ångström exponent (EAE)), the dust and other absorbing aerosols can be separated. However, it is difficult to separate BC and BrC based on the size information. To quantify the absorption contribution of BrC in the fine mode, a typical method was commonly used based on the strong spectral-dependence of BrC from UV to near-infrared

wavelengths. BrC is commonly seen to be non-absorbing in the near-infrared wavelengths, so the total fine aerosol absorption in near-infrared wavelengths comes completely from BC absorption (excluding dust). In UV wavelengths, the total absorption should be the sum of BrC and BC absorption, and the BrC absorption is the difference between the total absorption and BC absorption. Therefore, the derivation of the BrC absorption suffers large uncertainties from BC properties. The most widely used method to estimate the BrC absorption is based on the BC absorption Ångström exponent (AAE), which represents the

spectral dependence of the absorption. Given two wavelengths ($\lambda_1$ and $\lambda_2$), the BC AAE at the corresponding wavelength pair can be calculated using:

$$\mathrm{AAE} = -\frac{\ln(\frac{\mathrm{abs}(\lambda_1)}{\mathrm{abs}(\lambda_2)})}{\ln(\frac{\lambda_1}{\lambda_2})} \tag{1}$$

where $abs(\lambda_1)$ and $abs(\lambda_2)$ represent the absorptions at $\lambda_1$ and $\lambda_2$, respectively. Given the AAE value of BC, the BC absorption in UV wavelengths can be obtained based on the absorption in near-infrared wavelengths. However, there are large uncertainties

in the estimation of BC AAE. BC AAE = 1 is widely assumed, while the particle size, morphology, and mixing states have significant impacts on BC AAE values (Kirchstetter et al., 2004; Schnaiter et al., 2003; Li et al., 2016; Liu et al., 2018; Zhang

et al., 2020; Liu and Mishchenko, 2018). For example, for bare BC, Schnaiter et al. (2003) reported an average AAE value of approximately 1.1 for diesel BC aerosols; Kirchstetter et al. (2004) have shown BC AAE was approximately 0.6 – 1.3 for BC near the roadway or in the tunnel. Recent studies have realized that BC morphology, particle size, and mixing states can

lead to sizable uncertainties in BC AAE (Li et al., 2016; Lack and Cappa, 2010; Liu et al., 2018; Luo et al., 2020). A recent study conducted by Wang et al. (2016) used the Mie calculation to constrain the effects of particle size on the AAE, while a spherical BC morphology was assumed. In their study, a pre-calculated wavelength-dependence of AAE (WDA) based on Mie calculation was used, while the effects of BC morphology was not considered. In the atmosphere, BC presents rather complex morphologies based on the observation of electron microscopy images (China et al., 2013; Wang et al., 2017). To estimate

BrC absorption based on measurements from satellite or ground-based measurements, previous studies have developed some techniques to constrain the aerosol refractive index and aerosol type (Tesche et al., 2011; Arola et al., 2011). However, most studies have neglected the effects of BC morphologies. Even though recent studies have also shown that BC morphologies can affect BC AAE, few studies have provided direct evidence on how large deviations BC morphologies can cause for the estimation of BrC absorption.

As measurements in the atmosphere are caused by many factors including particle size, refractive index, mixing states, morphologies, etc., it is difficult to figure out how BC morphologies affect BrC absorption derivation. Besides, it is hard to quantify the deviations due to the effects of aerosol composition and size distributions (Li, Z. and Zhao, X. and Kahn, R. and Mishchenko, M. and Remer, L. and Lee, K.-H. and Wang, M. and Laszlo, I. and Nakajima, T. and Maring, H., 2009). As many studies have shown that BC complex morphologies can have an important impact on the BC optical properties, some studies

guessed that the AAE methods may not provide inaccurate estimations. However, few studies have provided direct evidence to prove their assumptions, and the simplified methods were still widely used. In many cases, we can expect that the simplified models may lead to deviations, but we cannot expect how large deviations the simplified models will cause. By using the real measurements, we cannot also expect under what circumstances the simplified models will lead to large deviations, and it is difficult to analyze how the deviations are caused.

Numerical tools have an edge on revealing the complex factors that affect the measurements and can be the supplements for the measurements. In this work, we replaced the complex measurements in the atmosphere with the well-constrained pseudo absorption "measurements" computed using morphologically realistic mixed models, and the inferred BrC absorptions based on the BC AAE = 1, Mie AAE, and Mie wavelength-dependent AAE methods were compared with the "True" BrC absorption. Also, the causes of the deviations were analyzed, and the method used in this work is shown in Figure 1. By using this inverse

framework, we intend to answer the following questions:

1. If BC presents a complex morphology, how large deviations in the estimation of BrC absorption the commonly used AAE methods will cause?

2. Under what conditions the simplified methods can provide bad/good estimations?

3. How the deviations between the True and the estimated BrC absorption using simplified models are caused?

## 2 Pseudo measurements

### 2.1 Morphologies

Non-spherical aerosol models show more excellent performance on reproducing the measurements even though the Mie theory was commonly used in remote sensing and climate modeling (Bi et al., 2018; He et al., 2016, 2015; Chakrabarty et al., 2007; Luo et al., 2019). In the atmosphere, BC can be mixed with BrC, and the mixing states are commonly divided into externally mixed and internally mixed. For the externally mixed particles, each chemical component is separated, and the BrC and BC absorption can be treated individually. However, in many cases, BC and BrC can be internally mixed. As BC is internally mixed with BrC, the total absorption can be enhanced by the "lensing effect "(Bond and Bergstrom, 2006; Lack et al., 2009) or weakened by the "sunglass effect" (Luo et al., 2018b).

The pseudo measured absorptions were calculated based on the morphologically realistic BC models. For the externally mixed particles, a fractal morphology was assumed for BC, and the structures satisfy the fractal law (Sorensen, 2001; Mishchenko et al., 2002):

$$N_s = k_0 (\frac{R_g}{R})^{D_f} \tag{2}$$

where $N_s$ and $R$ represent the monomer number and mean monomer radius, respectively; $D_f$ denotes the fractal dimension, and larger $D_f$ generally represents more compact aggregates. $k_0$ represents the fractal frefator, and it mainly affects the shape anisotropy. $R_g$ represents the gyration radius. To generate BC aggregates, a tunable algorithm was applied (Woźniak, 2012). In the tunable code, $k_0$ and $D_f$ are fully adjustable, and the fractal law is strictly satisfied in each growth step.

Bond and Bergstrom (2006) have demonstrated that the observed monomer radius was commonly in the range of 10 - 25 nm. However, previous studies have shown that the mass absorption cross-sections (MAC) of BC are insensitive to the BC monomer radius as BC monomer radius is in the range of 10 - 25 nm (Kahnert, 2010; Liu and Mishchenko, 2005). Therefore, similar to Luo et al. (2018b, c), a constant monomer radius of 20 nm was assumed. As both fluffy and compact BC exist in the atmosphere, we used $D_f$ = 1.8 and $D_f$ = 2.6 to represent the fluffy and compact BC, respectively. Even though the $k_0$ was also observed in a relatively wide range, when $D_f$ = 1.82, Liu and Mishchenko (2005) indicated that with $k_0$ increasing from approximately 0.9 to approximately 2.1, the BC MAC did not vary substantially. In this work, we fixed $k_0$ to be 1.2. According to Zhang et al. (2008), we used mobility diameters of 155 nm and 320 nm to represent small and large BC, respectively. As BC shape is irregular, we substituted the volume-mean BC diameter ($D_V = 2R(N_s)^{1/3}$) for the mobility diameter. The corresponding $N_s$ are 58 and 512, respectively. The morphology of externally mixed BrC was assumed to be spherical, as externally mixed BrC commonly exists as the spherical tarballs (Chakrabarty et al., 2010). BC refractive index can vary with wavelengths, while Bond and Bergstrom (2006) have shown that the BC refractive index does not vary largely with the wavelengths from ultraviolet to near-infrared region. In addition, this study mainly focuses on the effects of BC morphology, and the variation of BC refractive index is not considered, so we assumed a constant value for the BC refractive index. Bond and Bergstrom (2006) have suggested five values for the BC refractive indices, and we used the median value of 1.85 + 0.71i in this work. The real part of the BrC refractive index was assumed to be 1.55 (Chakrabarty et al., 2010).

For the internally mixing particles, the BC-containing morphologies were generated based on the models proposed by Luo et al. (2019). Here we simply describe the algorithm to generate the internally mixing particles. Firstly, we have generated the bare BC aggregates using the tunable code, and the bare BC aggregates were discretized into numerous dipoles. Then the coating materials were added based on two coating methods. The first coating method identifies the edge dipoles (BC surface) first, and then adds the coating materials based on a parameter $q$ (Luo et al., 2019):

$$q = \sum_{i=1}^{N_d} \frac{1}{L_i^k} \tag{3}$$

where $N_d$ is the number of edge dipoles, and $L_i$ represents the distance between an exterior dipole and the center of the $i$th edge dipole. The exterior dipole with a larger $q$ value is more easily filled with coating materials. $k$ is a tunable parameter, and with larger $k$, the coating materials are more easily to fill the dipoles surrounding the edge dipoles. In this study, $k = 8$ was assumed, and the generated BC model was referred to as Model A.

The second coating method adds the coating materials based on another parameter (Luo et al., 2019):

$$p = \sum_{i=1}^{N_c} \frac{1}{l_i^2} \tag{4}$$

where $N_c$ represents the BC monomer number within a defined sphere with a radius of $R_c$. The defined sphere can represent the uneven distribution of coating materials. $l_i$ represents the distance between an exterior dipole and the center of the $i$th monomer sphere. The exterior dipole with a larger $q$ value is assumed to be more easily filled with coating materials. In the second coating method, $R_c$ is assumed to be adjustable, and smaller $R_c$ can reflect more spherical coating materials. In this work, $R_c = 50R_g$ and $R_c = R_g$ were assumed to represent the film and spherical coatings, and are named Model B and Model C, respectively. In our previous study (Luo et al., 2019), we have demonstrated that our proposed models can greatly simulate the internally mixed BC morphologies and reproduce the measured absorption as well. For more details about the algorithm to generate the coated BC, please refer to Luo et al. (2019), and the typically generated morphologies are shown in Figure 3 and the Figures S1-S2 of Luo et al. (2019).

## 2.2 Generation of pseudo measurements

The Mie theory (Mie, 1908), the multiple-sphere T-matrix (MSTM) method (Mackowski and Mishchenko, 2011, 1996), and the discrete dipole approximation (DDA) method (Draine and Flatau, 2008, 1994) are widely used to calculate the optical properties of black carbon. The Mie theory is the most efficient method, but it is just applicable to spherical particles. MSTM was developed to calculate the optical properties of multiple spheres. Compared to DDA, it calculates analytically the optical properties of randomly oriented particles without numerically averaging over particle orientations. So, MSTM is more efficient and accurate than DDA. Bare BC is widely assumed to be composed of numerous spherical monomers, which can be calculated using the MSTM. Therefore, we used the MSTM to calculate the optical properties of bare BC, and MSTM version 3 was used in this work. However, as BC is coated with BrC, the mixed particle morphology becomes extremely complex, and it is difficult to fit the particle morphology using a group of spheres. DDA has an edge on calculating the optical properties of particles with

arbitrary shapes. Therefore, we used the DDA to calculate the optical properties of internally mixed particles. In this work, we used DDSCAT version 7.3 (Draine and Flatau, 2008, 1994). We assumed that BC is randomly orientated in the atmosphere (Mishchenko and Yurkin, 2017), and the results were averaged over $12 \times 7 \times 12 = 1080$ directions. In DDSCAT, the accuracy of the calculation depends significantly on the dipole spacing (d). In this work, all the calculations satisfy : $|m|k_W d < 0.23$, where m, $k_W$ are the refractive index of BC and wavenumber, respectively.

In MSTM and DDSCAT, the total absorption efficiency ($Q_{abs}$) of particles was directly outputted. In MSTM and DDSCAT, $Q_{abs}$ was defined with respect to the volume–mean radius, so the absorption cross-section ($C_{abs}$) can be obtained using:

$$C_{abs} = \frac{1}{4} Q_{abs} \pi D_V^2 \tag{5}$$

To verify the accuracy of MSTM and DDSCAT, we have compared the $C_{abs}$ of spherical BC calculated using the Mie theory, MSTM, and DDSCAT, respectively. The Mie calculations were performed using the PyMieScatt package (Sumlin et al., 2018). As shown in Figure 2, the $C_{abs}$ of spherical BC calculated using different numerical methods are in great agreement. The deviations between MSTM and Mie calculations are less than 0.1%. For bare BC, the deviations between DDSCAT and Mie calculations are less than 2%, and for core-shell BC, the deviations between DDSCAT and Mie calculations are less than 1%. The deviations are acceptable compared to the deviations between the "True" and inferred BrC absorption.

In real circumstances, the total absorptions can be inferred from the observations or measurements. Thus, the total absorption cross-section was used to provide pseudo measurements. For the internally mixed particles, the total absorption cross-section can be directly obtained from the calculations based on the morphologically realistic models. For the externally mixed particles, the total absorption cross-section is the sum of the absorption cross-section of BC and BrC.

## 3 Inferring the BrC absorption

### 3.1 "True" BrC absorption

In the study of Luo et al. (2018b), by separating the absorption of BC and BrC, they found the total absorption of the internally mixed particles can be less than the sum of BrC and BC absorption calculated individually. So, there must be a negative effect to weaken the total absorption. From physical points, Luo et al. (2018b) found that the BrC absorption can block the solar radiation deeply into BC, so weaken the total absorption, and the effect was named as "sunglass effect". In addition, the "lensing effect" was redefined as the absorption enhancements of BC by the addition of non-absorbing coating materials. Therefore, the total absorptions of mixed particles consist of BC absorption, BrC absorption, the "lensing effect, and the "sunglass effect". However, both the sunglass effect and BrC shell absorption are caused by absorbing BrC. For convenient application, the "True" BrC absorption was assumed as the difference between the absorption of BC mixed with BrC and BC mixed with non-absorbing materials. Here we must clarify that the "True" BrC absorption in this work is the co-effect of the absorption BrC shell and the "sunglass effect" for internally mixed particles. To eliminate the effect of BrC mass, the BrC mass absorption cross-section ($MAC_{BrC}$) was used, and it can be calculated using:

$$C_{abs\_BrC} = C_{abs\_BC \text{ and } BrC} - C_{abs\_BC \text{ and } non-absorbing} \tag{6}$$

$$MAC_{BrC} = C_{abs\_BrC}/M_{BrC} \tag{7}$$

here $C_{abs\_BC\ and\ BrC}$ and $C_{abs\_BC\ and\ non-absorbing}$ represent the absorption cross-sections of BC mixed with BrC and non-absorbing materials, respectively. The morphologies of BC mixed with non-absorbing materials is the same as those mixed with BrC ; $M_{BrC}$ represents the mass of BrC, which was calculated using:

$$M_{BrC} = V_{BrC} \cdot \rho_{BrC} \tag{8}$$

$$V_{BrC} = V_{BC} \cdot (1 - f_{BC})/f_{BC} \tag{9}$$

$$V_{BC} = N_s \cdot (4/3\pi R^3) \tag{10}$$

where $V_{BrC}$ and $V_{BC}$ represent the volume of BrC and BC, respectively; $f_{BC}$ represents the volume fraction of BC; $\rho_{BrC}$ represents the mass density of BrC. Even though the estimated BrC absorption cross-section is indepent of $\rho_{BrC}$, BrC MAC is significantly affected by $\rho_{BrC}$. We assumed that the BrC has the same mass density as the typical organic carbon (OC). However, the OC mass density ($\rho_{OC}$) varies in different regions. Even though Turpin and Lim (2001) suggested a typical value of 1.2 g/cm$^3$ for $\rho_{OC}$, they also observed a rather low $\rho_{OC}$ value of 0.87 g/cm$^3$. In addition, Turpin and Lim (2001) further showed that the reported $\rho_{OC}$ can vary from approximately 0.77 to approximately 1.9 g/cm$^3$. In this work, similar to Luo et al. (2018b), we just used the suggested value of 1.2 g/cm$^3$, and the uncertainties caused by $\rho_{OC}$ should be further evaluated in the future.

## 3.2 Inferring BrC absorption

The calculation of inferred BrC absorption is similar to the true case, while the difference is the $C_{abs\_BC\ and\ non-absorbing}$ is inferred from an assumed AAE:

$$C_{abs\_BC\_non-absorbing2} = C_{abs\_BC\_non-absorbing1} \cdot (\frac{\lambda_2}{\lambda_1})^{-AAE} \tag{11}$$

here $C_{abs\_BC\_non-absorbing1}$ and $C_{abs\_BC\_non-absorbing2}$ are the corresponding absorption cross-section of BC with non-absorbing materials at $\lambda_1$ and $\lambda_2$, respectively.

The total absorption observations at 440, 675, 870 nm wavelengths can be commonly obtained in AERONET and other ground measurements. Based on the strong spectral-dependence of BrC, BrC absorption at 675 and 870 nm wavelengths are commonly neglected, and the absorptions at 675 and 870 nm wavelengths come fully from the BC absorption. As BC absorption at 440 nm wavelength can be obtained based on the BC AAE, we can estimate the BrC absorption at 440 nm based

on Equation 6. In this work, we inferred the mass absorption cross-section (MAC) of BrC at 440 nm wavelength based on the pseudo measurements at 675 and 870 nm wavelength using BC AAE = 1 and AAE of Mie calculations. For the Mie AAE methods, we have pre-calculated the AAE of BC with a spherical structure (BC sphere or BC core-shell) by assuming an identical volume-mean diameter to the non-spherical BC using MSTM.

In addition, Wang et al. (2016) proposed a method to derive BrC absorption based on the AAE spectral-dependence (WDA) using Mie calculations. The WDA was calculated using:

$$\text{WDA} = \text{AAE}_{\lambda 1\_\lambda 2} - \text{AAE}_{\lambda 2\_\lambda 3} \tag{12}$$

where $\text{AAE}_{\lambda 1\_\lambda 2}$ and $\text{AAE}_{\lambda 2\_\lambda 3}$ are the AAE calculated based on different wavelength pairs. Based on the particle sizes and refractive index, the WDA was pre-calculated by assuming a spherical particle morphology, and then the AAE at a wavelength pair is inferred from AAE at another wavelength pair and pre-calculated WDA. As for the spherical BC, the optical properties are also calculated using MSTM but not the Mie method for convenience. However, the deviations between MSTM and Mie method for spherical BC are rather small, as shown in Figure 2. In this work, the WDA is calculated using MSTM by assuming a spherical morphology, and then the AAE between UV and near-infrared wavelengths are inferred from WDA and AAE in near-infrared wavelengths. Take the wavelengths of 440 nm, 675 nm, and 870 nm, for example, AAE between 440 nm and 675 nm can be calculated using:

$$\text{WDA} = \text{AAE}_{440nm\_870nm\_Mie} - \text{AAE}_{675nm\_870nm\_Mie} \tag{13}$$

$$\text{AAE}_{440nm\_870nm\_inferred} = \text{AAE}_{675nm\_870nm\_True} + \text{WDA} \tag{14}$$

where $\text{AAE}_{Mie}$ is the AAE of spherical BC with the same volume-mean diameter as the "True" case; $\text{AAE}_{inferred}$ and $\text{AAE}_{True}$ are the inferred BC AAE and the AAE calculated using the detailed BC models, respectively. For the inverse of BrC absorption, all the WDA was calculated based on the spherical BC by assuming an identical volume-mean diameter to the non-spherical BC, and we call it the WDA method. We have also demonstrated the effects of morphologies on the applicability of the WDA method. The "True" BrC absorption cross-section ($C_{abs\_BrC\_True}$) and the extimated BrC absorption cross-section ($C_{abs\_BrC\_Estimated}$) can be calculated using:

$$C_{abs\_BrC\_True} = C_{abs\_BC \text{ and } BrC} - C_{abs\_BC \text{ and } non-absorbing\_True} \tag{15}$$

$$C_{abs\_BrC\_Estimated} = C_{abs\_BC \text{ and } BrC} - C_{abs\_BC \text{ and } non-absorbing\_Estimated} \tag{16}$$

where $C_{abs\_BC \text{ and } non-absorbing\_True}$ and $C_{abs\_BC \text{ and } non-absorbing\_Estimated}$ represent the "True" and estimated absorption cross-section of BC mixed with non-absorbing materials, respectively.

As the BrC absorption estimation is significantly affected by the BC physical properties, we have also calculated the difference between "True" and the estimated BrC MAC:

$$\delta_{C_{abs}} = C_{abs\_BrC\_Estimated} - C_{abs\_BrC\_True} = C_{abs\_BC \text{ and } non-absorbing\_True} - C_{abs\_BC \text{ and } non-absorbing\_Estimated} \qquad (17)$$

Here we used a parameter $\delta_{MAC}$ to represent the difference of "True" and the estimated BrC absorption:

$$\delta_{MAC} = \delta_{C_{abs}}/M_{BrC} \qquad (18)$$

As the BrC MAC deviation between "Ture" and inferred BrC absorption is mainly caused by the inaccurate estimation of BC absorption, $\delta_{MAC}$ can represent the deviation between the "True" and inferred BrC MAC.

## 4  Results

### 4.1  Externally mixed particles

The BrC MAC is significantly depending on the imaginary part of the BrC refractive index. The measured imaginary parts of BrC refractive indices were varied largely in different pieces of literature. For example, Nakayama et al. (2013) showed that the secondary OC generated from the photooxidation of toluene has an imaginary part of refractive index from 0 to 0.0082 and from 0 – 0.0017 at 405 nm and 532 nm respectively; Saleh et al. (2013) showed that the imaginary part of primary OC refractive indices was in the range of 0.0055 – 0.06, while the imaginary parts of secondary OC refractive indices varied in the range of 0.01 – 0.05. Even though the imaginary part of BrC refractive index varies in different studies due to different chemical compositions, aging status, and generating process, the reported values were commonly within the range between the values reported by Kirchstetter et al. (2004) and those reported by Chen and Bond (2010). In general, the measured imaginary part of the BrC refractive index is commonly within the range of approximately 0 – 0.16.

The measured BrC MAC was also varied in different studies. The range of from 1.26 to 1.79 m$^2$/g at 365 nm wavelength was reported by previous studies (Cheng et al., 2011; Du et al., 2014; Srinivas et al., 2016), while Cho et al. (2019) reported a mean BrC MAC of approximately 0.7 m$^2$/g at 565 nm. BrC absorption properties based on laboratory measurements in urban and biomass smoke samples at Lawrence Berkeley National Laboratory showed BrC MAC of 2.75, 0.95, 0.42, 0.32, and 0.21 at $\lambda$= 400, 500, 600, 700, and 900 nm, respectively. In this work, the "True" BrC MAC is generally within the range of approximately 0 – 4m$^2$/g as the imaginary part of the BrC refractive index varies in the range of 0 – 0.16. Our calculated BrC mass absorption cross-section range is a little wider than the measurements as a wide imaginary part range of the BrC refractive index is assumed.

The comparisons of the "True" and inferred BrC absorption for externally mixed particles are shown in Figure 4. In general, the inferred BrC MAC agrees relatively well with the "True" BrC absorption when the BC fraction is small. This is easy to be understood. The total effects caused by the BC morphology are alleviated by the large BrC fraction, so the effects of BC morphology on the inferred mass BrC absorption is small. However, as the ratio of BC volume to BrC volume reaches 1:1,

the inferred BrC MAC based on the AAE methods may be significantly affected by the BC morphology. For the large particle, the Mie AAE methods may provide inaccurate estimations for both fluffy and compact particles, and the Mie AAE methods can overestimate the BrC mass absorption by approximately 4.8 $m^2$/g, which is approximately several times the observed BrC absorption. For small particles, the BrC absorption deviations estimated using the Mie AAE methods are relatively small for both fluffy and compact BC. As the morphological effects on the BrC absorption derivation are significantly dependent on the particle size, we have also investigated $\delta_{MAC}$ at different particle sizes. As shown in Figure 5, the accuracy of the Mie AAE method is significantly related to the particle size. Fixing $D_f$ to be 1.8, while the Mie AAE methods can provide a relatively reasonable estimation for small particles, the $\delta_{MAC}$ can increase with $D_V$, and it can reach approximately 4.8 $m^2$/g when the particle size is large. As shown in Figure 6, spherical BC AAE depends significantly on the particle size, and the AAE can reach a negative value for large BC. However, for fractal BC aggregates, the AAE is still around 1 even for large BC, so the Mie AAE methods provide rather inaccurate estimations for large particles.

The applicability of the BC AAE = 1 method should be also carefully considered. As freshly emitted BC commonly exhibits a near fluffy fractal structure (Chakrabarty et al., 2006; Wentzel et al., 2003; China et al., 2015), the AAE = 1 method can generally provide a reasonable estimation for BrC mixed with freshly emitted BC. As shown in Figure 4, fixing $D_f$ to be 1.8, the deviation between the "True" and the estimated BrC using the BC AAE = 1 method is not large. The reason is that the AAE of fluffy BC does not deviate largely with 1 (see Figure 6). However, the BC AAE = 1 method can provide less accurate estimations for BrC mixed with compact BC. Even though BrC MAC is relative accurately estimated for small particles, for the large particle, fixing $f_{BC}$=50%, most BrC mass absorption cross-section inferred by assuming AAE = 1 is below 0, and the underestimation of BrC MAC can reach approximately 2.3 $m^2$/g, as the AAE of large compact BC can be approximately 0.7 (Liu et al., 2018). Therefore, the BC AAE = 1 method is a reasonable method for freshly emitted particles, while it may provide rather inaccurate estimations for BrC mixed with compact BC aggregates.

To dispose of the effects of particle size on the AAE method, Wang et al. (2016) proposed a method based on the WDA method to derive BrC absorption. However, the WDA method does not necessarily provide a better estimation than using the Mie AAE and AAE = 1 methods, as the BC morphology in the atmosphere is rather complex. As shown in Figure 4, assuming that the BC morphology presents a fractal structure, the WDA method may provide worse estimations than using the BC AAE=1 method. As shown in Figure 5, the accuracy of the WDA method is significantly dependent on the particle size. As $D_V$ is approximately 100 nm, BrC MAC can be underestimated by approximately 9 $m^2$/g using the WDA method. Besides, the WDA method cannot provide a good estimation even for BrC mixed with compact BC, and for large particles, the WDA method can provide a worse estimation for BrC mixed with compact BC compared to BrC mixed with fluffy BC. To compare the WDA of spherical BC and fractal BC, we have calculated the WDA of fractal aggregates with $D_V$ varying from 40 to 400 nm based on the calculated database from our previous work (Luo et al., 2018a), where the BC refractive index was assumed to be $m = 1.95 + 0.79i$. As shown in Figure 6, the WDA of spherical BC depends largely on the particle size, while the WDA of fractal aggregates does not deviate largely with zero. Therefore, the effects of BC morphologies on the applicability of the WDA method should be carefully considered. Besides, we also notice that even though the "True" BrC absorption is larger

than 0, the inferred BrC absorption can be below 0 as the BC contents become large. Therefore, we should carefully consider the BC contents when using the AAE method to estimate the BrC absorption.

## 4.2 Internally mixed particles

As BC and BrC are internally mixed, the morphologies become more complex. Not only the fractal parameters (such as $D_f$) may change, but also the coating configurations may affect the morphologies. To demonstrate the effect of morphologies, we used three BC models based on different coating configurations to calculate the absorption of the internally mixed particles, as referred to above. As shown in Figure 7, different BrC coating shapes may lead to sizable variations in the "True" BrC absorption. Fixing BrC refractive index to be 1.55+0.08i, the variations in the BrC mass absorption cross-section caused by
different BrC coating shapes can vary from 0 to approximately 0.25 m$^2$/g. Besides, the particle size and compactness of mixed particles can also have significant effects on BrC absorption. Therefore, the determination of BrC absorption based on the modeling method should consider the variation of BrC coating shapes for internally mixed BrC even though most externally mixed BrC presents a near-spherical shape.

The estimated BrC MAC also deviates largely from the "True" BrC MAC for internally mixed particles. The Mie AAE
methods can just provide relatively reasonable estimations for relatively small particles, and for large particles, the inferred BrC MAC based on the Mie AAE methods even deviates more largely from "True" BrC MAC compared to the externally mixed particles. Fixing $N_s$ to be 512, and $D_f$ to be 1.8, the inferred BrC MAC using the Mie AAE at 440 and 870 nm wavelength pair can overestimate the "True" BrC MAC by approximately 5.8 m$^2$/g. As also shown in Figure 9, fixing $D_f$ to be 1.8, $\delta_{MAC}$ estimated based on the Mie AAE methods is relatively small when the particle is small, while it increases to
approximately 5.8 m$^2$/g when the $D_V$ of the mixed particles increases to approximately 400 nm. Furthermore, even for heavily coated BC ($f_{BC}$ = 10%), the Mie AAE method can overestimate the BrC MAC by approximately 1.0 m$^2$/g (see both Figure 8 and Figure 9), which is comparable to the BrC MAC. The Mie AAE method can provide inaccurate estimations even for BrC mixed with compact BC ($D_f$ = 2.6), and the deviation can reach approximately 2.8 m$^2$/g when $f_{BC}$ = 50%.

The BC AAE = 1 method seems to be still a reasonable method for internally mixed particles with a fluffy BC core. As
shown in Figure 9, fixing $D_f$ to be 1.8, $\delta_{MAC}$ is generally within -1 – 1.2 m$^2$/g, which is much smaller than $\delta_{MAC}$ estimated using the Mie AAE methods. However, the $\delta_{MAC}$ estimated using the AAE = 1 method can reach a value that is comparable to the BrC MAC, so it is non-negligible in the estimation of BrC absorption. Furthermore, as BC becomes compact, the AAE = 1 method may provide more inaccurate estimations, and it can underestimate the BrC MAC by 2 – 3 m$^2$/g. The possible reason may be that the AAE of more compact BC deviates substantially with 1 when BC size is large, as demonstrated in the study of
Liu et al. (2018). Most recent measurements have shown that the average $D_f$s of both bare and coated BC present a relatively small value. For example, China et al. (2013) found that the $D_f$ of ambient BC emitted from the wildfire was generally within the range of 1.75 – 1.9; China et al. (2014) demonstrated that BC $D_f$ in the freeway was in the range of 1.43 – 2.1. Yuan et al. (2019) have shown that the BC $D_f$ at a remote site in the Southeastern Tibetan Plateau was generally in the range of approximately 1.67 – 1.93; In the North China Plain, Wang et al. (2017) showed that BC $D_f$s at background sites, mountaintop,
urban, and tunnel were generally 1.8 – 2.16. With a fluffy BC structure, the AAE = 1 method seems still a reasonable method.

However, the $\delta_{\mathrm{MAC}}$ should also be noticed, as it can reach a value that is comparable to the BrC MAC for internally mixed particles. Besides, some near-spherical BC particles were also observed (eg. Lewis et al. (2009)), which should be carefully considered.

Sometimes, the WDA method may even provide worse estimations than the BC AAE = 1 and Mie AAE methods. Fixing $N_s$ to be 58 and $f_{\mathrm{BC}}$ to be 50%, the WDA method can overestimate BrC MAC by approximately 2 m$^2$/g, which is comparable to "True" BrC MAC. As shown in Figure 9, fixing $D_f$ to be 1.8, as the particle size of the mixed particles increases, $\delta_{\mathrm{MAC}}$ based on the WDA method increases firstly and then decreases. The WDA method can overestimate the BrC MAC by approximately 2.5 m$^2$/g when $D_V$ of the mixed particles is approximately 200 nm. The reason may be that the WDA calculated using the Mie method overestimates the effect of the BC size. As shown in Figure 10, even though the WDA of Model A does not deviate largely with 0, the WDA of the core-shell sphere model depends largely on the particle size. So the Mie WDA can overestimate the effects of the particle size, and the WDA method is obviously limited by the BC morphologies.

Even though the morphologically realistic models have not been used in the real cases, but based on the BC morphologies collected in the atmosphere, we believe that if we can know the detailed BC morphologies, we can improve the estimations. Some studies have been conducted to investigate the BC morphologies in different regions, which can provide information for the estimation of BrC absorption. For example, by exploring the three-dimensional (3D) electron tomography method, Adachi et al. (2007) have analyzed the morphological characteristics of BC. Based on the two-dimensional (2D) electron tomography image and fractal theory, China et al. (2013) have characterized the BC structures emitted from the wildfire. Wang et al. (2017) have investigated the BC morphologies at background sites, mountaintop, urban, and tunnel in North China. Besides, Yuan et al. (2019) have investigated the externally mixed and internally mixed BC at a remote site in the Southeastern Tibetan Plateau. However, we acknowledge that the measurements are still not enough now, and further measurements on the BC morphological information are required to improve the estimation. This study highlights the effects of BC morphology on the estimation of BrC absorption, which may further promote the measurements of complex BC morphologies in different regions. By conducting such measurements, we expect to obtain the percentages of different BC morphologies, and the optical properties will be calculated based on the "average" of different BC morphologies based on a probability distribution of different BC morphologies in a real case (Wu et al., 2020). In the future, we expect to use the measured BC morphological information in a real case, while this study focuses on theoretical investigations on the effect of BC particle morphology on the estimation of BrC absorption based on commonly used AAE methods.

## 5 Conclusions

Some previous studies have guessed that the AAE methods may not provide inaccurate estimations, but few studies have provided direct evidence to prove their guess. In this work, based on an inverse framework, we provide a relatively new insight to investigate the BC morphological effect on the estimation of BrC absorption. To focus on the effects of BC morphologies, pseudo measurements were generated based on some morphological mixed BC models, then the BrC absorption was inferred

based on the AAE method. Even though the "True" BrC absorption is within the measured range, the inferred BrC absorption is significantly affected by the BC morphologies.

By investigating the estimated BrC absorption at different parameters, we have demonstrated under what conditions the AAE methods can provide good/bad estimations. Freshly emitted BC commonly presents a fluffy structure, and its AAE does not deviate largely with 1, so the BC AAE = 1 method can provide reasonable estimations. For the internally mixed particles, as most recent studies have demonstrated that the $D_f$ of coated BC also exhibits a relatively small value, the BC AAE = 1 method is still a reasonable selection. However, the deviation between the "True" and the estimated BrC MAC should be also

carefully considered if BC exhibits a complex morphology, as sometimes the $\delta_{MAC}$ estimated using the BC AAE = 1 method can reach a value that is comparable to the "True" BrC MAC. The Mie AAE methods can just provide relatively reasonable estimations for small particles, and the BrC absorption deviations estimated using the Mie AAE methods are rather substantial for large particles. If the BC core still exhibits a fluffy structure, the deviation between the "True" and the estimated BrC MAC using the Mie AAE methods can reach 4.8 m$^2$/g and 5.8 m$^2$/g for large externally and internally mixed particles, respectively.

Even for compact BC core, the $\delta_{MAC}$ estimated using the Mie AAE methods can reach approximately 2.8 m$^2$/g for large particles. The WDA method does not necessarily improve the estimations. In many cases, the WDA method even provides a worse estimation than the AAE = 1 and Mie AAE methods, and the deviation of BrC MAC estimated using the WDA method can reach approximately 9 m$^2$/g for externally mixed particles. As recent studies have shown BC commonly exhibits a fluffy structure but not a spherical structure, the estimation of BrC absorption based on the AAE method should carefully consider

the effects of BC morphologies. Our findings can guide the use of different AAE methods.

    By comparing the AAE/WDA of spherical BC and detailed BC morphologically realistic models, we have provided explanations for why the good/bad estimations were caused. The AAE does not deviate largely with 1 if BC presents a fluffy fractal structure, while it varies largely with $D_V$ if BC exhibit a spherical structure, and the AAE value of spherical BC can vary from a negative value to approximately 1.4. Our results also show that the WDA of fluffy BC and spherical BC exhibit rather different

values. For both externally and internally mixed particles, the WDA does not deviate largely with 0 if the BC core presents a fluffy structure, while the WDA of spherical BC can vary largely with the particle size changing, and this may account for the inaccurate BrC absorption estimations using the WDA method. Our results can provide useful advice on analyzing why the deviation between the estimated BrC absorption based on AAE methods and direct measurements are caused.

*Data availability.* Our calculations were performed using MSTM Version 3.0 and DDSCAT 7.3. MSTM Version 3.0 can be found online
(http://www.eng.auburn.edu/∼dmckwski/scatcodes/, ), and DDSCAT 7.3 can be obtained from http://ddscat.wikidot.com/. Our calculation results can be downloaded from: https://figshare.com/articles/dataset/Deriving_BrC_Revised_zip/12839570/2 (Luo, 2020).

*Author contributions.* JL and QZ conceived the research idea. JL performed the computations, and wrote the paper. YZ verified the simulation methods and results. QZ reviewed the paper and supervised the findings of this work. All authors discussed the results and contributed to the final paper.

*Competing interests.* The authors declare that no competing interests.

*Acknowledgements.* We thank the fund support of the National Natural Science Foundation of China (Grant No. 41675024 and U1733126); Fundamental Research Funds for the Central Universities (Grant No. WK2320000040). We particularly thank Dr. D. W. Mackowski and Dr. M. I. Mishchenko for the MSTM code, and thank Bruce Draine and Pjotr Flatau for the DDSCAT software. We also acknowledge the support of super computing center of University of Scenice and Technology of China.

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

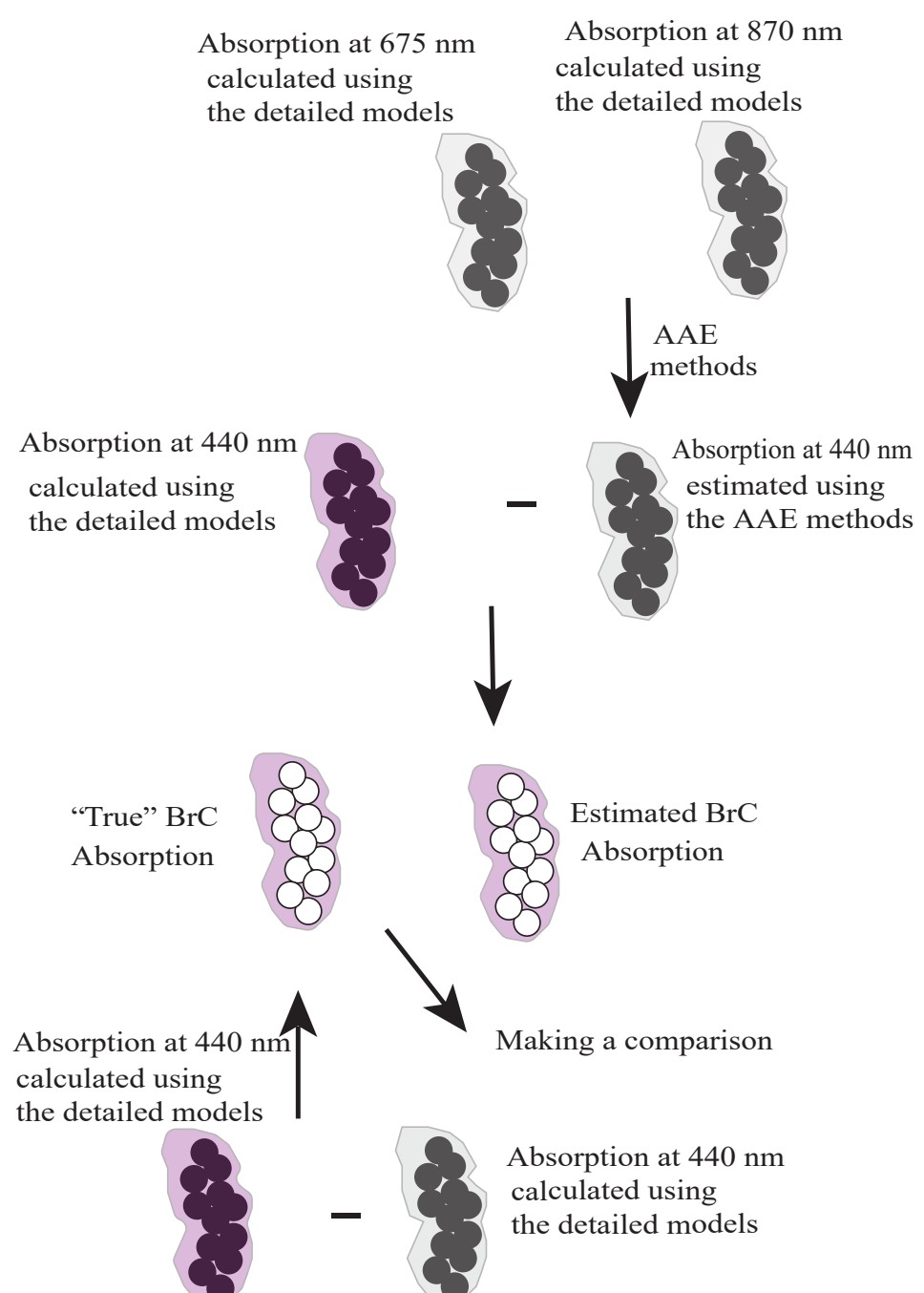

**Figure 1.** The estimation of BrC absorption.

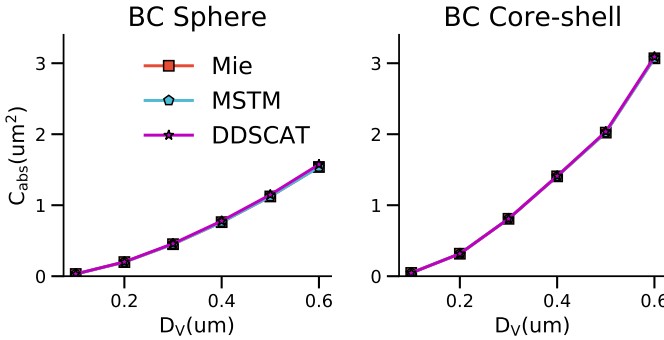

**Figure 2.** The absorption cross-sections of spherical BC calculated using the Mie theory, MSTM and respectively. For the BC core sphere, the ratio of the shell radius to the core radius was assumed to be 1.5.

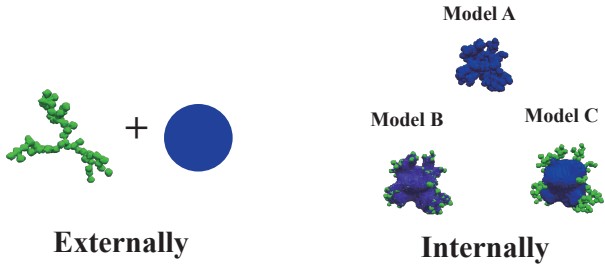

**Figure 3.** BC morphologies considered in this work. The internally mixed particles were generated using the models developed by Luo et al. (2019).

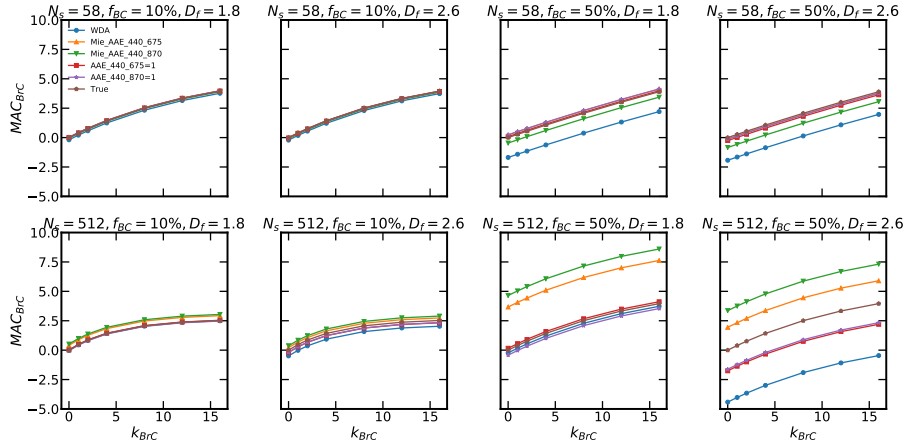

**Figure 4.** Comparison of the "True" and inferred BrC MAC, $\lambda = 440$ nm.

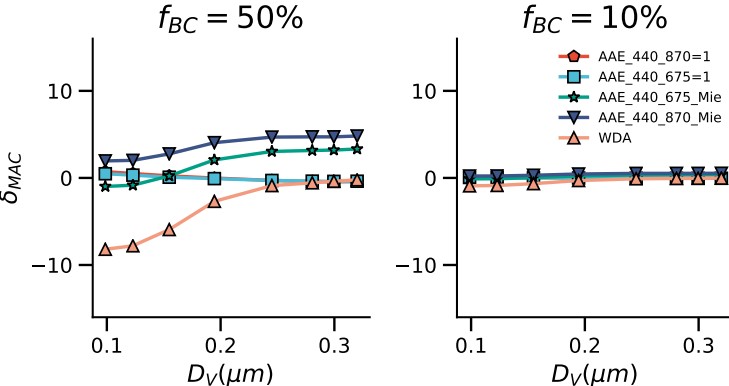

**Figure 5.** $\delta_{\mathrm{MAC}}$ of inferred and "Ture" BrC absorption, and here $D_V$ represents the equivalent volume size of BC, $\lambda = 440$ nm, $D_f$=1.8.

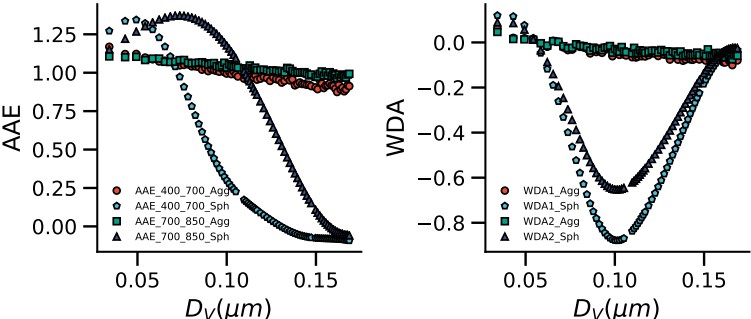

**Figure 6.** Comparison of AAE and WDA between BC sphere and aggregates ($D_f$=1.8, $m = 1.95 + 0.79i$). Here WDA1 represents the AAE difference between the 400 - 700 nm wavelength pair and the 700 - 850 nm wavelength pair; WDA2 represents the difference between the 400 - 850 nm wavelength pair and the 700 - 850 nm wavelength pair. "Agg" and "Sph" denote the fractal BC aggregates and spherical BC, respectively.

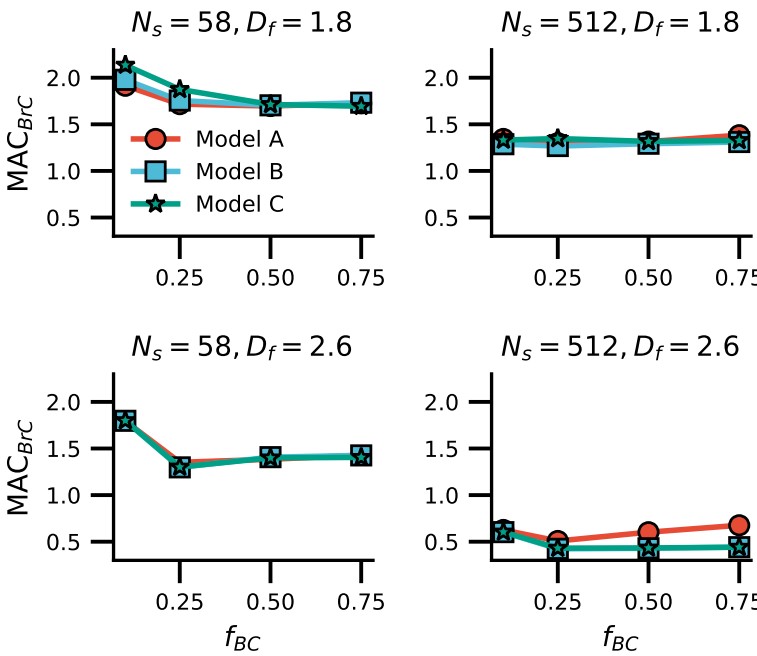

**Figure 7.** Variation of the "True" BrC absorption with different coating models, $k_{BrC}$=0.08, $\lambda = 440$ nm.

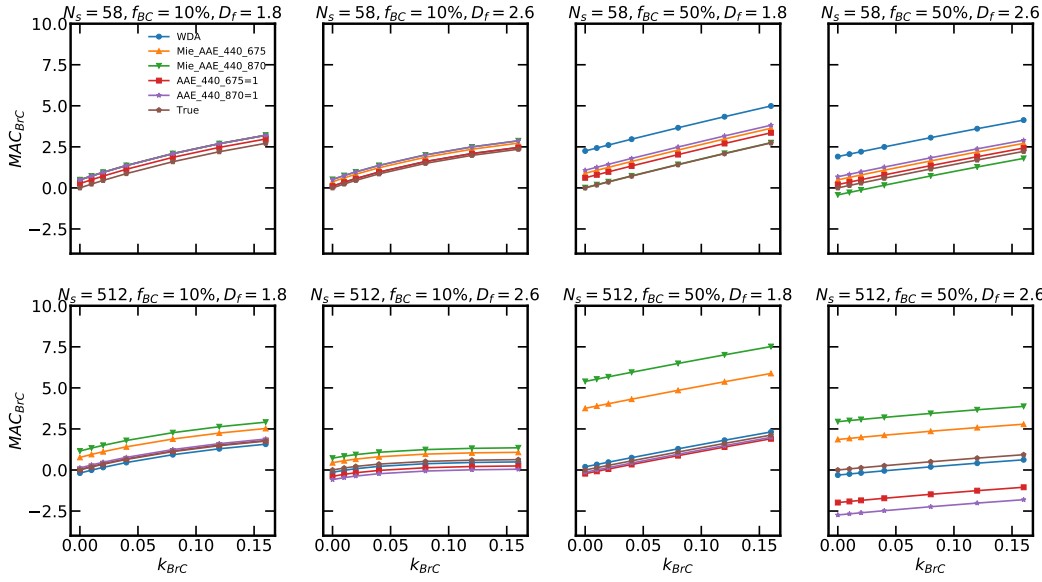

**Figure 8.** Comparison of the "True" and inferred BrC absorption for internally mixed particles (Model A), $\lambda = 440$ nm.

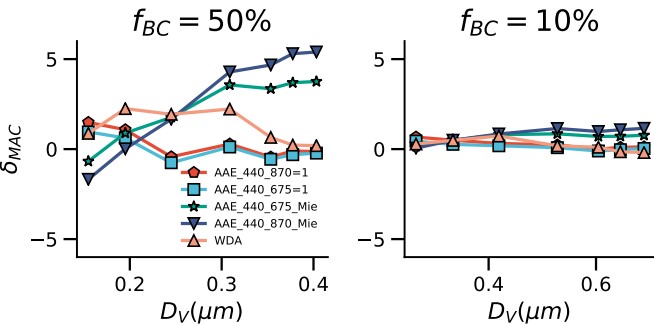

**Figure 9.** Similar to Figure 5, but for internally mixed particles ($D_f = 1.8$), $\lambda = 440$ nm.

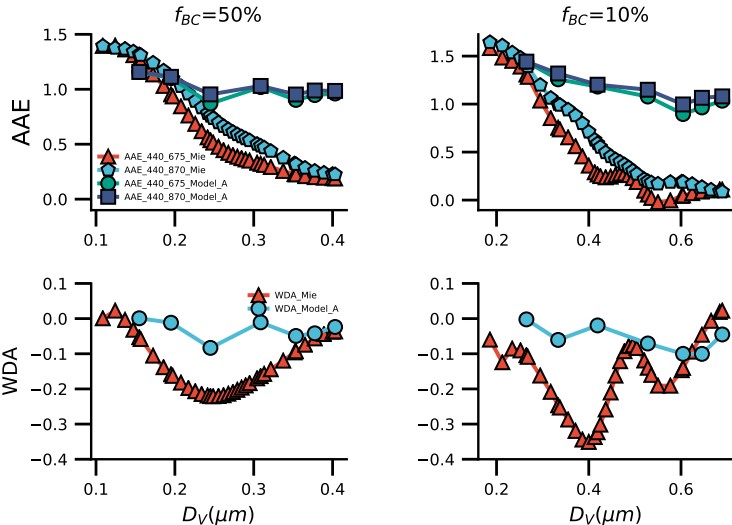

**Figure 10.** Comparison of AAE and WDA between core-shell sphere model and Model A. Here WDA represents the AAE difference between the 440 - 675 nm wavelength pair and the 440 - 870 nm wavelength pair.