# Peer review of "Effects of black carbon morphology on brown carbon absorption estimation: from numerical aspects"

_Geoscientific Model Development, 2020_

## Short Comment (SC1) · 14 Nov 2020

Dear authors,

in my role as Executive editor of GMD, I would like to bring to your attention our Editorial version 1.2:

https://www.geosci-model-dev.net/12/2215/2019/

and especially to Appendix A clarifying the code and data availability issue. Section A1 reads:

*Every paper must include a section at the end of the paper before the "Acknowledge-*

[Figure]

*ments" entitled "Code availability" or "Code and data availability" as appropriate.*

- *This section must include citations for the persistent public archives of the precise versions of all of the code and data associated with the paper. The generic means to access other versions of the code and data as well as the licence of the code should also be explained. The licence should conform to the Open Source Definition16. Suitable licences17 are for example GPL18 or MIT19.*

- *Where the authors cannot, for reasons beyond their control, publicly archive part or all of the code and data associated with a paper, they must clearly state the restrictions. They must also provide confidential access to the code and data for the editor and reviewers in order to enable peer review. The arrangements for this access must not compromise the anonymity of the reviewers. All manuscripts which do not make code and data available at this level are to be rejected. Where only part of the code or data is subject to these restrictions, the remaining code and/or data must still be publicly archived. In particular, authors must make every endeavour to publish any code whose development is described in the manuscript.*

*Code and data access must be provided at the time that the discussion paper is submitted. Embargoes, whether pending acceptance or for a defined period, are not acceptable.*

As you, for sure done some calculations including some kind of software. It is integral part of a GMD publication to include permanent access to these. Please make them available as soon as possible.

Yours,

Astrid Kerkweg

---

## Author Comment (AC1) · 23 Nov 2020

Dear Astrid Kerkweg,

Thanks for your comments. Our calculations were performed using MSTM Version 3.0 and DDSCAT 7.3. MSTM Version 3.0 can be found online (http://www.eng.auburn.edu/~dmckwski/scatcodes/), and DDSCAT 7.3 can be obtained from http://ddscat.wikidot.com/. Our calculation results can be downloaded from https://figshare.com/articles/dataset/Deriving_BrC_Revised_zip/12839570. We will add the details in the code and data vailability section. Thank you!

Kind regards,

Qixing Zhang on behalf of the coauthors

---

## Referee Comment (RC1) · Anonymous Referee #1 · 26 Nov 2020

The authors conducted theoretical calculations to quantify the effect of BC particle morphology on inferring brown carbon absorption based on three commonly used AAE methods. The BC morphology issue has been investigated a lot in the past 10 years, particularly for its impact on BC absorption. This study provides a relatively new perspective to look at the BC morphology effect on deriving brown carbon absorption through spectral/AAE methods. The implication for the advantages and disadvantages of those common AAE methods could be important to guide future measurements and retrieval of BrC absorption. However, the presentation in a number of places in the text is quite confusing to me and requires further clarification and more explanations, particularly in the methodology part. Please see my specific comments below.

[Figure]

Major Comments:

1. Section 2.1: More descriptions are needed for the algorithm and model used to generate bare BC aggregates and coated BC particles. At least the key steps and elements involved in the algorithm and model need to be presented in addition to simply citing the references.

2. Equation (3): I am not quite convinced that this is the best way to compute the absorption cross section of BC with irregular shapes. Would it be better to use the projected area (averaged cross all directions) than pi/4 * Dvˆ2 (volume-equivalent geometric cross section)? Besides, can MSTM and DDA methods directly output the absorption cross sections? If so, why did the authors need to use equation (3)? Based on lines 109-110, it seems that DDA can directly compute absorption cross section for the entire particle with irregular shapes. Why not using DDA for both external and internal mixing cases? Did the MSTM and DDA can produce exactly the same results for the same case? If not, then using two different methods could further introduce differences between external and internal results.

3. Lines 137-147: This part is not clear to me. How could delta_MAC represent the deviation between the "True" and inferred BrC MAC? What if this delta_MAC can be affected by the additional absorption from BrC, which interacts with BC physical properties? Currently, delta_MAC is only calculated from the difference between "True" and the estimated BC absorption by assuming BC is mixed with non-absorbing materials. Why not directly compute the difference between "True" and the estimated absorption for BC mixed with BrC?

4. Section 3.2: The way to infer BrC absorption is also not very clear to me. For example, (1) Line 157, "estimate the BrC absorption at 440 nm based on Equation 1", should it be based on Equation (4)? (2) Line 159: "AAE of Mie calculation". Could the authors be more specific about how did they compute this AAE using Mie calculation? Assuming core-shell structure for BC coated by BrC? (3) How did the delta_MAC in

Equation (10) fit into the analysis?

5. A number of assumptions used in this study could affect the results and conclusions. For example, the assumed BrC density, how much uncertainty would this bring into the final results?

Minor Comments:

1. The language needs to be further polished particularly to correct grammatic issues. Just to name a few: Line 31: "divide BC and BrC" should be "separate BC and BrC". Line 34: "exclude the dust" should be "excluding dust". Line 65: it should be "BC AAE", right? etc. I suggest the authors carefully check the entire text again.

2. Could the authors give some comments on how their results/conclusions could help future measurements of BrC absorption?

---

## Referee Comment (RC2) · Anonymous Referee #2 · 15 Dec 2020

This paper presents results from numerical experiments aimed at exploring the bias in quantifying BrC absorption based on methods that do not account for complex BC morphology. The experiments involve constructing BC/BrC particles with complex morphologies and employing 3 AAE attribution methods to retrieve BrC absorption. As expected, the numerical experiments show deviation between "true" and retrieved BrC absorption.

Major comments:

1) The fact that AAE attribution methods or assuming spherical particles (i.e. Mie calculations) lead to bias in attributing measured absorption to BC and BrC is well
known. However, there is a reason why these approximations are employed, which is that it is not possible (or at least not feasible) to account for the complex mixing state and morphology of atmospheric aerosols. It is not clear that utilizing a detailed optical model (e.g. DDA) that does not accurately represent the mixing state and morphology would do enough of a better job than simplified (e.g. Mie) calculations to justify its use in interpreting ambient observations and/or in large-scale models.

Put in other words, utilizing detailed optical models requires single-particle level knowledge of mixing state and morphology, which is not currently feasible. The examples given in the paragraph starting at Line 272 are not enough to provide such information. If one uses these types of measurements, they'll end up with a situation where they have to assume some sort of "average" population morphology based on a parameterization similar to Equation (3) – it is not clear that such an exercise would substantially improve the representation of reality compared to the AAE attribution or Mie calculations.

The results in this paper (e.g. in Figure 3) are specific to the parameters employed in the numerical experiments and cannot be generalized in any quantitative sense to help interpret real measurements. Therefore, the paper in its current form does not tell us more than what we already know, which is that simplified methods can lead to uncertainty in BrC / BC absorption. For this paper to make an impact, it needs to make a convincing case that employing their complex morphologies in atmospheric retrievals provides improvements compared to the simplified methods. Or it needs to provide useful information that can be utilized to quantify the uncertainty associated with the simplified methods.

2) Along the same lines, there is no justification as to the choice of "true" mixing states / morphologies and how representative they are of atmospheric aerosols. To play devil's advocate, for some of the "true" mixing states / morphologies, there is very good agreement between the true and simplified cases – what if those are the most representative of atmospheric aerosols? In this case, sticking to the current simplified approaches is

good enough.

3) The choice of optical models in the study is not well justified. Why is MSTM used for externally mixed particles and DDA for internally mixed particles? For the reader to be convinced of the validity of the outputs of these models, they need to be validated against each other and against Mie calculations. The authors need to show that for spherical homogeneous particles and spherical core-shell particles, MSTM, DDA, and Mie calculations yield the same absorption cross-sections.

4) For this paper to be publishable, it requires substantial language editing. There are many instances of misuse of articles, inappropriate word choice, and incoherent sentence structure.

Specific comments:

1) Figure 1: This figure is confusing. As referenced in the text, it is supposed to be an overview of the method, but I don't think it accomplishes this goal.

2) Line 21: estimation of BrC what?

3) Line 75: well mixed is not appropriate here as it usually refers to the case where the components do not form separate phases. Since lensing is mentioned, here the authors refer to core-shell morphology, which has 2 separate phases and is not well mixed.

4) Line 165-167: MSTM and Mie calculations should produce the exact same results for spherical particles. This should actually be done to validate MSTM – if it deviates from Mie for spherical, the results for non-spherical cannot be trusted.

5) Line 174: Here you mention that WDA is calculated based on Mie theory, but earlier (Line 166) you state that MSTM was used in the calculations.

6) Line 183: this is not uncertainty (though uncertainty exists) but real variability in optical properties due to variability is BrC chemical composition. Using "suffer" is not

appropriate here since it is actual variability.

---

## Author Comment (AC2) · 30 Jan 2021

The comment was uploaded in the form of a supplement:
https://gmd.copernicus.org/preprints/gmd-2020-348/gmd-2020-348-AC2-supplement.pdf

---

## Author Response (AR1)

**Responses to the comments of reviewers**

(The responses are highlighted in blue)

The authors really appreciate the valuable comments and constructive suggestions from the reviewers. The questions and comments of reviewers are in black font, and responses are highlighted in blue. The changes made in the revised manuscript are marked in red.

**Response to the comments of Reviewer # 1**

The authors conducted theoretical calculations to quantify the effect of BC particle morphology on inferring brown carbon absorption based on three commonly used AAE methods. The BC morphology issue has been investigated a lot in the past 10 years, particularly for its impact on BC absorption. This study provides a relatively new perspective to look at the BC morphology effect on deriving brown carbon absorption through spectral/AAE methods. The implication for the advantages and disadvantages of those common AAE methods could be important to guide future measurements and retrieval of BrC absorption. However, the presentation in a number of places in the text is quite confusing to me and requires further clarification and more explanations, particularly in the methodology part. Please see my specific comments below.

**Response:** Thanks very much for your comments, the specific responses are shown in the following parts.

Major Comments:

1. Section 2.1: More descriptions are needed for the algorithm and model used to generate bare BC aggregates and coated BC particles. At least the key steps and elements involved in the algorithm and model need to be presented in addition to simply citing the references.

**Response:** Thanks very much for your comments. We are very sorry for without clarifying the algorithm and model used to generate bare BC aggregates and coated BC particles. We have added some simple and key descriptions in the revised manuscript. Please see the section 2.1 in the revised manuscript.

2. Equation (3): I am not quite convinced that this is the best way to compute the absorption cross section of BC with irregular shapes. Would it be better to use the projected area (averaged cross all directions) than pi/4 * Dv^2 (volume-equivalent geometric cross section)? Besides, can MSTM and DDA methods directly output the absorption cross sections? If so, why did the authors need to use equation (3)? Based on lines 109-110, it seems that DDA can directly compute absorption cross section for the entire particle with irregular shapes. Why not using DDA for both external and

internal mixing cases? Did the MSTM and DDA can produce exactly the same results for the same case? If not, then using two different methods could further introduce differences between external and internal results.

**Response:** Thanks very much for your comments. We are sorry for without clarifying the output of MSTM and DDSCAT. MSTM and DDSCAT can calculate the absorption cross-sections, but directly output a total absorption efficiency ($Q_{abs}$). Both in MSTM and DDSCAT, $Q_{abs}$ is defined as:

$$Q_{abs} = C_{abs}/\pi a_{eff}^2$$

where $a_{eff}$ is the "effective radius", and it is represented by the radius of an equal volume sphere in MSTM and DDSCAT. Therefore, we used equation (3).

As shown in Figure 1, the calculations of MSTM and DDSCAT for the spherical BC are in great agreement. Besides, Luo et al. (2019) have demonstrated that the calculations of MSTM and DDSCAT for the bare BC aggregates and closed-cell BC model are also in good agreement, as shown in Figure 2.

[Figure]

Figure 1 The comparison of Mie, MSTM, and DDSCAT for the Spherical BC, λ= 440 nm.

[Figure]

Figure 2 The comparison of MSTM and DDSCAT for the BC aggregates and closed-cell model, where λ= 532 nm, and the monomer number is 58. This Figure is replotted from the results of Luo et al. (2019), where $D_p/D_c$ represents the ratio of the total monomer particle diameter to the BC core dimeter.

3. Lines 137-147: This part is not clear to me. How could delta_MAC represent the deviation

between the "True" and inferred BrC MAC? What if this delta_MAC can be affected by the additional absorption from BrC, which interacts with BC physical properties? Currently, delta_MAC is only calculated from the difference between "True" and the estimated BC absorption by assuming BC is mixed with non-absorbing materials. Why not directly compute the difference between "True" and the estimated absorption for BC mixed with BrC?

**Response:** Thanks very much for your comments. delta_MAC is also equal to the difference between "True" and the estimated absorption for BC mixed with BrC. In principle, delta_MAC should be calculated by the difference between "True" and the estimated absorption for BC mixed with BrC. However, the "True" BrC absorption was calculated using:

$$C_{abs\_BrC\_True} = C_{abs\_BC\_and\_BrC} - C_{abs\,BC\,and\_non\_absorbing\_True}$$

$$C_{abs\_BrC\_Estimated} = C_{abs\_BC\_and\_BrC} - C_{abs\,BC\,and\_non\_absorbing\_Estimated}$$

Therefore, the difference between the estimated BrC absorption and "True" and can be calculated using:

$$\delta_{C_{abs}} = C_{abs\_BrC\_Estimated} - C_{abs\_BrC\_True} = C_{abs\,BC\,and\_non\_absorbing\_True}$$

$$- C_{abs\,BC\,and\_non\_absorbing\_Estimated}$$

Therefore,

$$\delta_{MAC} = (C_{abs\,BC\,and\_non\_absorbing\_True} - C_{abs\,BC\,and\_non\_absorbing\_Estimated})/m_{BrC}$$

We have clarified it in the revised manuscript.

4.     Section 3.2:    The way to infer BrC absorption is also not very clear to me.    For example, (1) Line 157, "estimate the BrC absorption at 440 nm based on Equation 1",should it be based on Equation (4)? (2) Line 159: "AAE of Mie calculation". Could the authors be more specific about how did they compute this AAE using Mie calculation? Assuming core-shell structure for BC coated by BrC?    (3) How did the delta_MAC Equation (10) fit into the analysis?

**Response:** Thanks very much for your comments. We are very sorry for without clarifying clearly the method to infer BrC absorption. It is indeed true that here it should be Equation (4). We are very sorry for the mistakes, and we have modified it in the revised manuscript.

For the "AAE of Mie calculation", we used the MSTM to calculate the AAE of BC by assuming a spherical structure. We have clarified it in the revised manuscript:

"For the Mie AAE methods, we have pre-calculated the AAE of BC with a spherical structure (BC

sphere or BC core-shell) by assuming an identical volume-mean diameter to the non-spherical BC."

"Based on the particle sizes and refractive index, the WDA was pre-calculated by assuming a spherical particle morphology, and then the AAE at a wavelength pair is inferred from AAE at another wavelength pair and pre-calculated WDA."

Since the results of MSTM and Mie calculation are in good agreement, it is acceptable to replace Mie with MSTM. We have compared the results of Mie (performed using the PyMieScatt package), MSTM, and DDSCAT for spherical BC, as shown in Figure 1. We found that the deviation between MSTM and Mie is less than 0.1%, so it is acceptable to use MSTM for convenience.

As for how the delta_MAC Equation (10) fits into the analysis, please refer the last response.

5. A number of assumptions used in this study could affect the results and conclusions. For example, the assumed BrC density, how much uncertainty would this bring into the final results?

**Response:** Thanks very much for your comments. We indeed made some assumptions, and these assumptions may lead to some uncertainties. However, as we mainly focus on the effects of BC morphology, these assumptions can just affect the specific values but don't significantly affect the general conclusions. We have added some comments in the revised manuscript.

The BrC density was assumed to be 1.2 $g/cm^3$, but the values may vary in different regions. Even though Turpin and Lim (2001) suggested a typical value (1.2 $g/cm^3$) for OC mass density, a lower OC mass density of 0.87 $g/cm^3$ was also observed at a background site. In addition, Turpin and Lim (2001) also showed that the reported OC mass density can vary from approximately 0.77 to 1.9 $g/cm^3$. In this work, we just used the typical value (1.2 $g/cm^3$) suggested by Turpin and Lim (2001), and the uncertainties caused by the OC mass density should be further evaluated in the future. As the absorption cross-section deviation caused by the BC morphologies is independent of the BrC density, the BrC density just affects the BrC MAC. We have clarified it in the revised manuscript:

"Even though the estimated BrC absorption cross-section is independent of $\rho_{BrC}$, BrC MAC is significantly affected by $\rho_{BrC}$. We assumed that the BrC has the same mass density as the typical organic carbon (OC). However, OC mass density ($\rho_{OC}$) varies in different regions. Even though Turpin and Lim (2001) suggested a typical value of 1.2 $g/cm^3$ for OC mass density, they also observed a rather low $\rho_{OC}$ value of 0.87 $g/cm^3$. In addition, Turpin and Lim (2001) further showed that the reported $\rho_{OC}$ can vary from approximately 0.77 to approximately 1.9 $g/cm^3$. Similar to Luo et al. (2018b), we just used the suggested value of 1.2 $g/cm^3$, and the uncertainties caused by the OC mass density should be further evaluated in the future."

In addition, the BC refractive index was assumed to be a constant value of 1.85+0.71i, while some

studies have shown that it depends on wavelengths. However, Bond and Bergstrom (2006) also showed that the BC refractive index does not vary significantly from ultraviolet to near-infrared region, and they have suggested five values for the BC refractive indices. We used the median value of 1.85+0.71i. In addition, this works mainly focuses on the effects of BC morphology, and and the effect of BC refractive index is beyond the scope of our research. We have also clarified it in the revised manuscript:

"BC refractive index can vary with wavelengths, while Bond and Bergstrom (2006) have shown that the BC refractive index does not vary largely with the wavelengths from ultraviolet to near-infrared region. In addition, this study mainly focuses on the effects of BC morphology, and the variation of BC refractive index is not considered, so we assumed a constant value for the BC refractive index. Bond and Bergstrom (2006) have suggested five values for the BC refractive indices, we used the median value of $1.85 + 0.71i$ in this work."

Minor Comments:

1. The language needs to be further polished particularly to correct grammatic issues. Just to name a few: Line 31: "divide BC and BrC" should be "separate BC and BrC". Line 34: "exclude the dust" should be "excluding dust". Line 65: it should be "BC AAE", right? etc. I suggest the authors carefully check the entire text again.

**Response:** Thanks very much for your comments. We have checked the English carefully in the revised manuscript, and the modifications were marked in red in the revised manuscript.

2. Could the authors give some comments on how their results/conclusions could help future measurements of BrC absorption?

**Response:** Thanks very much for your comments. We have rewritten the conclusion section and it includes some comments on how our results/conclusions could help future measurements of BrC absorption.

Firstly, we have pointed out the advantage/disadvantages of different AAE methods, which can guide the use of different AAE methods:

"By investigating the estimated BrC absorption at different parameters, we have demonstrated in which conditions the AAE methods can provide good/bad estimations. Freshly emitted BC commonly presents a fluffy structure, and its AAE does not deviate largely with 1, so the AAE = 1 method can provide reasonable estimations. For the internally mixed particles, as most recent studies have demonstrated that the $D_f$ of coated BC still exhibits a relatively small value, the AAE = 1 method is still a reasonable selection. However, the deviation between the "True" and the estimated

BrC MAC should be also carefully considered if BC exhibits a complex morphology, as sometimes the $\delta_{MAC}$ estimated using the BC AAE = 1 method can reach a value that is comparable to the "True" BrC MAC. The Mie AAE method can just provide relatively reasonable estimations for small particles, and the BrC absorption deviations estimated using the Mie AAE methods are rather substantial for large particles. If the BC core is still a fluffy structure, the deviation between the "True" and the estimated BrC MAC can reach 4.8 $m^2$/g and 5.8 $m^2$/g for large externally and internally mixed particles, respectively. Even for compact BC core, the $\delta_{MAC}$ estimated using the Mie AAE method can reach approximately 2.8 $m^2$/g for large particles. The WDA method does not necessarily improve the estimations. In many cases, the WDA method even provides a worse estimation than the AAE = 1 and Mie AAE methods, and the deviation of BrC MAC estimated using the WDA method can reach approximately 9 $m^2$/g for externally mixed particles. As recent studies have shown BC commonly exhibits a fluffy structure but not a spherical structure, the estimation of BrC absorption based on the AAE method should carefully consider the effects of BC morphologies. Our findings can guide the use of different AAE methods."

In addition, by comparing the AAE/WDA of spherical BC and detailed BC models, we have provided the explanations for why the good/bad estimations were caused. Our findings can provide useful advice on analyzing why the deviation between the estimated BrC absorption based on AAE methods and direct measurements are caused. We have rewritten the conclusions:

"By comparing the AAE/WDA of spherical BC and detailed BC morphologically realistic models, we have provided the explanations for why the good/bad estimations were caused. The AAE does not deviate largely with 1 if BC presents a fluffy fractal structure, while it varies largely with $D_V$ if BC exhibit a spherical structure, and the AAE value of spherical BC can vary from a negative value to approximately 1.4. Our results also show that the WDA of fluffy BC and spherical BC exhibit rather different values. For both externally and internally mixed particles, the WDA does not deviate largely with 0 if the BC core presents a fluffy structure, while the WDA of spherical BC can vary largely with the particle size changing, and this may account for the inaccurate BrC absorption estimations using the WDA method. Our results can provide useful advice on analyzing why the deviation between the estimated BrC absorption based on AAE methods and direct measurements are caused."

**Response to the comments of Reviewer # 2**

This paper presents results from numerical experiments aimed at exploring the bias in quantifying BrC absorption based on methods that do not account for complex BC morphology. The

experiments involve constructing BC/BrC particles with complex morphologies and employing 3 AAE attribution methods to retrieve BrC absorption. As expected, the numerical experiments show deviation between "true" and retrieved BrC absorption.

**Response:** Thanks very much for your comments, the specific responses are shown in the following parts.

Major comments:

1) The fact that AAE attribution methods or assuming spherical particles (i.e. Mie calculations) lead to bias in attributing measured absorption to BC and BrC is well known. However, there is a reason why these approximations are employed, which is that it is not possible (or at least not feasible) to account for the complex mixing state and morphology of atmospheric aerosols. It is not clear that utilizing a detailed optical model (e.g. DDA) that does not accurately represent the mixing state and morphology would do enough of a better job than simplified (e.g. Mie) calculations to justify its use in interpreting ambient observations and/or in large-scale models.

Put in other words, utilizing detailed optical models requires single-particle level knowledge of mixing state and morphology, which is not currently feasible. The examples given in the paragraph starting at Line 272 are not enough to provide such information. If one uses these types of measurements, they'll end up with a situation where they have to assume some sort of "average" population morphology based on a parameterization similar to Equation (3) – it is not clear that such an exercise would substantially improve the representation of reality compared to the AAE attribution or Mie calculations.

The results in this paper (e.g. in Figure 3) are specific to the parameters employed in the numerical experiments and cannot be generalized in any quantitative sense to help interpret real measurements. Therefore, the paper in its current form does not tell us more than what we already know, which is that simplified methods can lead to uncertainty in BrC / BC absorption. For this paper to make an impact, it needs to make a convincing case that employing their complex morphologies in atmospheric retrievals provides improvements compared to the simplified methods. Or it needs to provide useful information that can be utilized to quantify the uncertainty associated with the simplified methods.

**Response:** Thanks very much for your comments. Our responses are shown in the following aspects:

**(1) Does this study just provide information that we already know?**

To the best of our knowledge, previous studies have not provided direct evidence showing that BC

complex morphologies have a significant impact on the BrC absorption. As many studies have shown that BC complex morphologies can have an important impact on the BC optical properties, some studies guessed that the AAE methods may not provide inaccurate estimations, but the simplified methods were still widely used. In many cases, we can expect that the simplified models may lead to deviations, but we cannot expect how large deviations the simplified models will cause, and under what conditions the simplified models will lead to large deviations, and we cannot analyze how the deviations are caused. Based on an inverse framework, we provide a relatively new insight to investigate the BC morphological effect on the estimation of BrC absorption. By performing such theoretical calculations, we can directly see under what conditions that the simplified models will provide bad/good estimations, and we have analyzed reasons for the deviations. Therefore, we think that this work can improve the understanding of the deviation in the estimation of BrC absorption, and provide an implication for the advantages and disadvantages of the commonly used AAE methods. We have clarified it in the revised manuscript:

"As many studies have shown that BC complex morphologies can have an important impact on the BC optical properties, some studies guessed that the AAE methods may not provide inaccurate estimations. However, few studies have provided direct evidence to prove their assumptions, and the simplified methods were still widely used. In many cases, we can expect that the simplified models may lead to deviations, but we cannot expect how large deviations the simplified models can cause. By using the real measurements, we cannot also expect under what conditions the simplified models can lead to large deviations, and it is difficult to analyze how the deviations are caused."

**(2)     What implications this work can provide for atmospheric science?**

Firstly, we have pointed out the advantage/disadvantages of different AAE methods. By investigating the estimated BrC absorption at different parameters, we have demonstrated under what conditions the AAE methods can provide good/bad estimations, which can guide the use of different AAE methods:

"By investigating the estimated BrC absorption at different parameters, we have demonstrated under what conditions the AAE methods can provide good/bad estimations. Freshly emitted BC commonly presents a fluffy structure, and its AAE does not deviate largely with 1, so the AAE = 1 method can provide reasonable estimations. For the internally mixed particles, as most recent studies have demonstrated that the $D_f$ of coated BC still exhibits a relatively small value, the AAE = 1 method is still a reasonable selection. However, the deviation between the "True" and the estimated BrC MAC should be also carefully considered if BC exhibits a complex morphology, as sometimes

the $\delta_{MAC}$ estimated using the BC AAE = 1 method can reach a value that is comparable to the "True" BrC MAC. The Mie AAE method can just provide relatively reasonable estimations for small particles, and the BrC absorption deviations estimated using the Mie AAE methods are rather substantial for large particles. If the BC core is still a fluffy structure, the deviation between the "True" and the estimated BrC MAC can reach 4.8 m$^2$/g and 5.8 m$^2$/g for large externally and internally mixed particles, respectively. Even for compact BC core, the $\delta_{MAC}$ estimated using the Mie AAE method can reach approximately 2.8 m$^2$/g for large particles. The WDA method does not necessarily improve the estimations. In many cases, the WDA method even provides a worse estimation than the AAE = 1 and Mie AAE methods, and the deviation of BrC MAC estimated using the WDA method can reach approximately 9 m$^2$/g for externally mixed particles. As recent studies have shown BC commonly exhibits a fluffy structure but not a spherical structure, the estimation of BrC absorption based on the AAE method should carefully consider the effects of BC morphologies. Our findings can guide the use of different AAE methods."

In addition, by comparing the AAE/WDA of spherical BC and detailed BC models, we have provided the explanations for why the good/bad estimations were caused. This can provide useful advice on analyzing why the deviations between the estimated BrC absorption based on AAE methods and direct measurements are caused. We have rewritten the conclusions:

"By comparing the AAE/WDA of spherical BC and detailed BC morphologically realistic models, we have provided the explanations for why the good/bad estimations were caused. The AAE does not deviate largely with 1 if BC presents a fluffy fractal structure, while it varies largely with $D_V$ if BC exhibit a spherical structure, and the AAE value of spherical BC can vary from a negative value to approximately 1.4. Our results also show that the WDA of fluffy BC and spherical BC exhibit rather different values. For both externally and internally mixed particles, the WDA does not deviate largely with 0 if the BC core presents a fluffy structure, while the WDA of spherical BC can vary largely with the particle size changing, and this may account for the inaccurate BrC absorption estimations using the WDA method. Our results can provide advice on analyzing why the deviation between the estimated BrC absorption based on AAE methods and direct measurements are caused."

Our results show that sometimes the variation caused by the BC morphologies may even larger than the BrC absorption itself, so the BC morphology can significantly affect the estimation of BrC absorption.

Lastly, by comparing the deviations between the simplified models and detailed optical models, we can know how large deviations between the estimated BrC absorption using different AAE methods and the absorption of BrC mixed with BC with a complex morphology, which can provide useful

information for the analysis of the uncertainties. This work found that the BC morphology can significantly affect the estimation of BrC absorption, which may promote the measurements of BC morphological information.

**(3) Are the detailed models worse than the simplified models?**

It is indeed not clear now that whether the estimations can be improved by using a single detailed model, but based on the BC morphologies collected in the atmosphere, we believe that if we can know the detailed BC morphologies, we can improve the estimations. Some studies have been conducted to investigate the BC morphologies in different regions. For example, Adachi et al. (2007) have analyzed the morphological characteristics of BC. Based on the two-dimensional (2D) electron tomography image and fractal theory, China et al. (2013) have characterized the BC structures emitted from the wildfire. Wang et al. (2017a) have investigated the BC morphologies at background sites, mountaintop, urban, and tunnel in North China. Besides, Yuan et al. (2019) have investigated the externally mixed and internally mixed BC at a remote site in the Southeastern Tibetan Plateau. These measurements can provide morphological information for the estimation of BrC absorption.

But this is not enough, as BC morphologies are various in the atmosphere, so much more measurements are needed. However, to the best of our knowledge, just rather limited groups have conducted measurements on the BC morphologies. We think one of the problems is that rather limited studies have realized that BC morphologies can have a significant impact on the estimation of BrC absorption. Researchers just guess that BC morphologies may introduce errors in the BrC absorption estimation, but don't know how large deviations the simplified models will cause. The results of this work tell us that sometimes BC morphologies can introduce large errors which may even larger than several times of BrC absorption itself. Therefore, our study can raise attention to the effects of BC morphology on the estimation of BrC absorption.

Therefore, this study will also promote the measurements of complex BC morphologies in different regions. By conducting such measurements, we can know the percentages of different BC morphologies. In this work, we have provided examples for different BC morphologies, but in the real case, the optical properties will be calculated using the "average" of different BC morphologies based on a probability distribution of different BC morphologies, such as the study of Wu et al. (2020). In the future, we intend to use the detailed morphological models in a real case to see how the detailed morphological models can improve the estimation. However, this work mainly focuses on how the BC morphology affects the estimation of BrC absorption, and it can provide implications for the advantages/disadvantages of simplified models, which can provide some

guidance for the use of simplified models.

The focus of this work is not to use a detailed optical model in a real case. Instead, as black carbon exhibits a very complex morphology in the atmosphere, we intend to answer the following questions: if black carbon presents a complex morphology, how can the simplified models provide bad/good estimations? Why the deviations are caused? We have added some comments in the revised manuscript:

"Even though the morphologically realistic models have not been used in the real cases, but based on the BC morphologies collected in the atmosphere, we believe that if we can know the detailed BC morphologies, we can improve the estimations. Some studies have been conducted to investigate the BC morphologies in different regions, which can provide information for the estimation of BrC absorption. For example, by exploring the three-dimensional (3D) electron tomography method, Adachi et al. (2007) have analyzed the morphological characteristics of BC. Based on the two-dimensional (2D) electron tomography image and fractal theory, China et al. (2013) have characterized the BC structures emitted from the wildfire. Wang et al. (2017a) have investigated the BC morphologies at background sites, mountaintop, urban, and tunnel in North China. Besides, Yuan et al. (2019) have investigated the externally mixed and internally mixed BC at a remote site in the Southeastern Tibetan Plateau. However, we acknowledge that the measurements are still not enough now, and further measurements on the BC morphological information are required to improve the estimation. This study highlights the effects of BC morphology on the estimation of BrC absorption, which may further promote the measurements of complex BC morphologies in different regions. By conducting such measurements, we expect to obtain the percentages of different BC morphologies, and the optical properties will be calculated based on the "average" of different BC morphologies based on a probability distribution of different BC morphologies in a real case (Wu et al., 2020). In the future, we expect to use the measured BC morphological information in a real case, while this study focuses on theoretical investigations on the effect of BC particle morphology on the estimation of BrC absorption based on commonly used AAE methods. "

**References:**

Adachi, K., Chung, S. H., Friedrich, H., and Buseck, P. R.: Fractal parameters of individual soot particles determined using electron tomography: Implications for optical properties, Journal of Geophysical Research: Atmospheres, 112, 2007

China, S., Mazzoleni, C., Gorkowski, K., Aiken, A. C., and Dubey, M. K.: Morphology and mixing state of individual freshly emitted wildfire carbonaceous particles, Nature Communications, 4,

<GotoISI>://WOS:000323715900002, 2013.

China, S., Salvadori, N., and Mazzoleni, C.: Effect of Traffic and Driving Characteristics on Morphology of Atmospheric Soot Particles at Freeway On-Ramps, Environmental Science and Technology, 48, 3128–3135, <GotoISI>://WOS:000333776000007, 2014.

Wu, Y., Cheng, T., and Zheng, L.: Light absorption of black carbon aerosols strongly influenced by particle morphology distribution, Environ Res Lett, 15, 094051, 10.1088/1748-9326/aba2ff, 2020.

Wang, Y., Liu, F., He, C., Bi, L., Cheng, T., Wang, Z., Zhang, H., Zhang, X., Shi, Z., and Li, W.: Fractal dimensions and mixing structures of soot particles during atmospheric processing, Environmental Science & Technology Letters, 4, 487–493, 2017a

Yuan, Q., Xu, J., Wang, Y., Zhang, X., Pang, Y., Liu, L., Bi, L., Kang, S., and Li, W.: Mixing state and fractal dimension of soot particles at a remote site in the southeastern Tibetan plateau, Environmental science & technology, 53, 8227–8234, 2019.

2) Along the same lines, there is no justification as to the choice of "true" mixing states /morphologies and how representative they are of atmospheric aerosols. To play devil's advocate, for some of the "true" mixing states / morphologies, there is very good agreement between the true and simplified cases – what if those are the most representative of atmospheric aerosols?   In this case, sticking to the current simplified approaches is good enough.

**Response:** Thanks very much for your comments. To the best of our knowledge, previous studies have not provided direct evidence showing the impact of BC complex morphologies on the BrC absorption. After careful consideration, we also think that it is not reasonable to give up the simplified methods. Our results can provide implications for the advantage/disadvantage of simplified methods. So we have modified the conclusion in the revised manuscript:

"By investigating the estimated BrC absorption at different parameters, we have demonstrated under what conditions the AAE methods can provide good/bad estimations. Freshly emitted BC commonly presents a fluffy structure, and its AAE does not deviate largely with 1, so the AAE = 1 method can provide reasonable estimations. For the internally mixed particles, as most recent studies have demonstrated that the coated BC still exhibits a relatively small value, the AAE = 1 method is still a reasonable selection. However, the deviation between the "True" and the estimated BrC MAC should be also carefully considered if BC exhibits a complex morphology, as sometimes the $\delta_{MAC}$ estimated using the BC AAE = 1 method can reach a value that is comparable to the "True" BrC MAC. The Mie AAE method can just provide relatively reasonable estimations for small particles, and the deviations estimated using the Mie AAE methods are rather substantial for large particles.

If the BC core still exhibits a fluffy structure, the deviation between the "True" and the estimated BrC MAC can reach 4.8 m$^2$/g and 5.8 m$^2$/g for large externally and internally mixed particles, respectively. Even for compact BC core, the $\delta_{MAC}$ estimated using the Mie AAE method can reach approximately 2.8 m$^2$/g for large particles. The WDA method does not necessarily improve the estimations. In many cases, the WDA method even provides a worse estimation than the AAE = 1 and Mie AAE methods, and the deviation of BrC MAC estimated using the WDA method can reach approximately 9 m$^2$/g for externally mixed particles. As recent studies have shown BC commonly exhibits a fluffy structure but not a spherical structure, the estimation of BrC absorption based on the AAE method should carefully consider the effects of BC morphologies. Our findings can guide the use of different AAE methods.

By comparing the AAE/WDA of spherical BC and detailed BC models, we have provided explanations for why the good/bad estimations were caused. The AAE does not deviate largely with 1 if BC presents a fluffy fractal structure, while it varies largely with $D_V$ if BC exhibit a spherical structure, and the AAE value of spherical BC can vary from a negative value to approximately 1.4. Our results also show that the WDA of fluffy BC and spherical BC exhibit rather different values. For both externally and internally mixed particles, the WDA does not deviate largely with 0 if the BC core presents a fluffy structure, while the WDA of spherical BC can vary largely with the particle size changing, and this may account for the inaccurate BrC absorption estimations using the WDA method. Our results can provide advice on analyzing why the deviation between the estimated BrC absorption based on AAE methods and direct measurements are caused."

3) The choice of optical models in the study is not well justified. Why is MSTM used for externally mixed particles and DDA for internally mixed particles? For the reader to be convinced of the validity of the outputs of these models, they need to be validated against each other and against Mie calculations. The authors need to show that for spherical homogeneous particles and spherical core-shell particles, MSTM, DDA, and Mie calculations yield the same absorption cross-sections.

**Response:** Thanks very much for your comments. We are very sorry for without clarifying the choice of optical models. Mie, MSTM, and DDA are widely used to calculate the optical properties of black carbon. The Mie theory is the most efficient method, while it is just applicable for spherical particles. MSTM was developed to calculate the optical properties of multiple spheres. Compared to DDA, MSTM calculates analytically the optical properties of randomly oriented particles without numerically averaging over particle orientations. Consequently, MSTM is more efficient and accurate than DDA. Bare BC is widely assumed to be composed of spherical monomers, which can be calculated using the MSTM. Therefore, we used the MSTM to calculate the optical properties of

bare BC. However, as BC is coated with BrC, the mixed particle morphology becomes extremely complex, and it is difficult to reflect the particle morphology using a group of spheres. DDA has an edge on calculating the optical properties of particles with arbitrary shapes. Therefore, we used the DDA to calculate the optical properties of internally mixed particles.

We have compared the optical properties of spherical BC calculated using Mie, MSTM, and DDSCAT, respectively. We used the PyMiescatt software to conduct the Mie calculation. As shown in Figure 1, the deviations between MSTM and Mie calculations are less than 1%. For bare BC, the deviations between DDSCAT and Mie calculations are less than 2%, and for core-shell BC, the deviations between DDSCAT and Mie calculations are less than 1%. The deviations are acceptable compared to the deviations between the "True" and inferred BrC absorption. For the bare BC aggregates and the BC closed cell model, Luo et al. (2019) have shown that the results of DDSCAT and MSTM are in good agreement with the results of MSTM. Therefore, the deviations caused by different numerical methods are acceptable.

[Figure]

Figure 1 The comparison of Mie, MSTM, and DDSCAT for the Spherical BC.

4) For this paper to be publishable, it requires substantial language editing. There are many instances of misuse of articles, inappropriate word choice, and incoherent sentence structure.

**Response:** Thanks very much for your comments. We have checked carefully the English in the revised manuscript, and the modifications are marked in the revised manuscript.

Specific comments:

1) Figure 1: This figure is confusing. As referenced in the text, it is supposed to be an overview of the method, but I don't think it accomplishes this goal.

**Response:** Thanks very much for your comments. We have clarified that the "True" absorption is calculated using the detailed models in the revised Figure. The revised Figure is shown in Figure 2.

[Figure]

Figure 2 The revised Figure 1 in the revised manuscript (The estimation of BrC absorption.)

2) Line 21: estimation of BrC what?

**Response:** Thanks very much for your comments. We are very sorry for the mistakes. Here should be "estimation of BrC absorption".

3) Line 75: well mixed is not appropriate here as it usually refers to the case where the components do not form separate phases. Since lensing is mentioned, here the authors refer to core-shell

morphology, which has 2 separate phases and is not well mixed.

**Response:** Thanks very much for your comments. We have revised it accordingly in the revised manuscript.

4) Line 165-167:   MSTM and Mie calculations should produce the exact same results for spherical particles.   This should actually be done to validate MSTM – if it deviates from Mie for spherical, the results for non-spherical cannot be trusted.

**Response:** Thanks very much for your comments. We have validated the MSTM in the revised manuscript. As shown in Figure 1, the deviations between MSTM and Mie calculations are less than 1%. For bare BC, the deviations between DDSCAT and Mie calculations are less than 2%, and for core-shell BC, the deviations between DDSCAT and Mie calculations are less than 1%. The deviations are acceptable compared to the deviations between the "True" and inferred BrC absorption.

5) Line 174: Here you mention that WDA is calculated based on Mie theory, but earlier (Line 166) you state that MSTM was used in the calculations.

**Response:** Thanks very much for your comments. For spherical BC, the calculations using MSTM and Mie are in great agreement. So to ensure consistency, we used the MSTM calculations to replace the Mie calculations. We have clarified it in the revised manuscript:

"For the Mie AAE methods, we have pre-calculated the AAE of BC with a spherical structure (BC sphere or BC core-shell) by assuming an identical volume-mean diameter to the non-spherical BC using MSTM."

6) Line 183:   this is not uncertainty (though uncertainty exists) but real variability in optical properties due to variability is BrC chemical composition.   Using "suffer" is not appropriate here since it is actual variability.

**Response:** Thanks very much for your comments. We have modified the sentence as "Even though the imaginary part of BrC refractive index varies in different studies due to different chemical compositions, aging status, and generating process."